# Interaction of Proteins with a Planar Poly(acrylic acid) Brush: Analysis by Quartz Crystal Microbalance with Dissipation Monitoring (QCM-D)

**DOI:** 10.3390/polym13010122

**Published:** 2020-12-30

**Authors:** Jacek Walkowiak, Michael Gradzielski, Stefan Zauscher, Matthias Ballauff

**Affiliations:** 1Aachen-Maastricht Institute for Biobased Materials (AMIBM), Maastricht University, Brightlands Chemelot Campus, Urmonderbaan 22, 6167 RD Geleen, The Netherlands; j.walkowiak@maastrichtuniversity.nl; 2Stranski Laboratorium für Physikalische Chemie und Theoretische Chemie, Institut für Chemie, Straße des 17. Juni 124, Sekr. TC7, Technische Universität Berlin, 10623 Berlin, Germany; michael.gradzielski@tu-berlin.de; 3Mechanical Engineering and Material Sciences, Duke University, Durham, NC 27708, USA; 4Institut für Chemie und Biochemie, Freie Universität Berlin, Takustraße 3, 14195 Berlin, Germany

**Keywords:** polyelectrolyte brush, protein, ATRP, QCM-D

## Abstract

We describe the preparation of a poly(acrylic acid) (PAA) brush, polymerized by atom transfer radical polymerization (ATRP) of *tert*-butyl acrylate (*t*BA) and subsequent acid hydrolysis, on the flat gold surfaces of quartz-crystal microbalance (QCM) crystals. The PAA brushes were characterized by Fourier transform infrared (FT-IR) spectroscopy, ellipsometry and water contact angle analysis. The interaction of the PAA brushes with human serum albumin (HSA) was studied for a range of ionic strengths and pH conditions by quartz-crystal microbalance with dissipation monitoring (QCM-D). The quantitative analysis showed a strong adsorption of protein molecules onto the PAA brush. By increasing the ionic strength, we were able to release a fraction of the initially bound HSA molecules. This finding highlights the importance of counterions in the polyelectrolyte-mediated protein adsorption/desorption. A comparison with recent calorimetric studies related to the binding of HSA to polyelectrolytes allowed us to fully analyze the QCM data based on the results of the thermodynamic analysis of the binding process.

## 1. Introduction

Polymer brushes consist of polymer chains densely grafted by one end to an interface [1]. Interactions of proteins with such structures have been a subject of significant interest and investigation in colloid and polymer science for quite some time [1,2,3,4,5,6,7]. In many cases polymer brushes are studied to control the protein adsorption onto surfaces [8,9]. This becomes more complex when the brush is composed of charged polymer chains, i.e., polyelectrolytes [4,10,11,12,13,14]. Surfaces modified with polyelectrolyte brushes have been frequently investigated as they are related to “smart” or stimuli responsive surface coatings [15] and biosensors [16]. Furthermore, polyelectrolyte brushes can be applied to prevent biofouling [11]. It is known that proteins adsorbed and immobilized onto polyelectrolyte brushes can retain their conformation [17] as well as their (enzymatic) activity [18,19]. Understanding protein adsorption to polyelectrolyte brushes is therefore obviously needed for nanotoxicology and nanomedicine [20].

Proteins can be taken up or released from polyelectrolyte brushes depending on salt concentrations (see Figure 1) [21,22]. This is due to the counterion release mechanism [23] which is the main driving force for protein adsorption onto polyelectrolyte brushes [4]. Here the charged patches on the surface of the protein may function as multivalent counterions for the polyelectrolyte chain thus releasing a number of its condensed counterions. The gain in entropy thus achieved is the major driving force for the adsorption of proteins to polyelectrolytes. Therefore the effective degree of ionization and the charge distribution in the polyelectrolyte brush ─which strongly depends on the salt concentration and pH of the solution [24]—is crucial for protein adsorption.

The characterization of polyelectrolyte brushes and their stimulus response to changes in salt concentration and pH has been the focal point of a large number of articles. Thus, the effect of grafting density on brush conformation [25,26], the hysteretic memory of brushes [24], the ion specific effects on brush conformation [27] and the interactions with proteins [11,25,28] have all been studied in detail. We recently showed that proteins can adsorb onto like-charged spherical polyelectrolyte brushes (SPBs) [21,29,30]. Using FT-IR spectroscopy we established that the secondary structure of the adsorbed proteins was preserved [17,31]. Moreover, using small angle X-ray and neutron scattering (SAXS and SANS) we found that proteins are evenly distributed within the brush [29]. By coupled spectroscopic ellipsometry and quartz-crystal microbalance with dissipation monitoring (QCM-D), Bittrich et al. studied the amount of buffer component coupled to a PAA brush in swelling and protein adsorption process [32]. QCM-D is a technique that is sensitive to the mass change and the viscoelastic properties of an adsorbed layer [33,34]. Delcroix et al. studied the pH- and salt- dependent polymer conformation and protein adsorption on several polymer brushes [35]. In particular, they defined a protocol for systematic polymer conformational change evaluation upon protein adsorption by QCM-D.

Here we present a QCM-D study of HSA adsorption onto a planar PAA brush that allows us a quantitative comparison with our recent calorimetric studies of the same problem [30,36]. In this way precise structural information can be combined with thermodynamic information. We synthesized the brush layer by atom transfer radical polymerization (ATRP) [37,38,39]. The ATRP synthesis of PAA brushes consists of two steps, namely, the growth of P*t*BA brush and its subsequent acidic hydrolysis into PAA brush [40,41,42]. Surface-initiated ATRP (SI-ATRP) as a grafting from method for surface functionalization with polymer brushes was established recently by several groups [43,44]. Yadav et al. have grown P*t*BA brushes from silicon wafers using SI-ATRP and converted them into PAA brushes by acid hydrolysis [24]. Li et al. have grown mixed P*t*BA/polystyrene (PS) brushes from silica particles using living radical polymerization techniques with subsequent hydrolysis of P*t*BA towards PAA/PS brushes [45]. Borisova et al. showed that PAA brushes can be grafted from gold QCM crystals by nitroxide-mediated radical polymerization (NMP) [46]. Here we utilized an activator regenerated by electron transfer (ARGET) ATRP with the surface initiated procedure where an ATRP initiator is at first immobilized on the surface of a solid substrate [47]. The ARGET ATRP enables to drive the polymerization reaction to high conversion [48,49].

Here we also used the QCM-D technique to study polyelectrolyte brush swelling and protein adsorption to verify the utility of our synthetic route in functionalization of QCM crystals (see Figure 1). By analysis of QCM-D data based on thermodynamic study of a well-controlled model system [30,36] we present a comprehensive study on brush synthesis and its interaction with proteins.

The paper is organized as follows: in Section 2 we present the materials and experimental methods used for the synthesis and characterization of PAA brush. We describe the conditions of the QCM-D experiments and we present a short description of the methods used for data analysis. In the last section we present our results along with discussion devoted to PAA brush conformation and its interaction with HSA as a function of ionic strength (I) and pH. The comparison with calorimetric studies [30] allows us to analyze the QCM data on the basis of thermodynamic studies.

## 2. Materials and Methods

### 2.1. Materials

*tert-*Butyl acrylate (98%, Aldrich, Darmstadt, Germany) was destabilized by passing through a column of activated basic alumina. CuBr_2_ (98%, Aldrich, Darmstadt, Germany), *N*,*N*,*N*′,*N*′,*N*″–pentamethyldiethylenetriamine (PMDETA, 99%, Aldrich, Darmstadt, Germany), acetone (99.8%, Aldrich, Darmstadt, Germany), L-ascorbic acid (99%, Aldrich, Darmstadt, Germany), sodium dodecyl sulfate (SDS, 98.5%, Aldrich, Darmstadt, Germany) anhydrous dichloromethane (99.8%, Merck, Darmstadt, Germany), trifluoroacetic acid (99%, Aldrich, Darmstadt, Germany), 3-morpholinopropane-1-sulfonic acid (MOPS, 99.5%, Aldrich, Darmstadt, Germany) and human serum albumin (HSA, Mw: 66.5 kDa, 97%, Aldrich, Darmstadt, Germany) were used as received. The water was purified by filtration thorough a Millipore system (Barnstead, Lake Balboa, CA, USA) resulting in resistivity higher than 18 MΩcm. Gold-coated QCM crystals (5 MHz, 14 mm) were purchased from Q-Sense (Biolin Scientific, Västra Frölunda, Sweden) and cleaned prior to use. The synthesis and deposition of the surface-bound initiator bis[2-(2-bromoisobutyryloxy)undecyl]disulfide (DTBU) followed the published procedure [27].

### 2.2. Instrumentation

ATR-FTIR spectra were recorded using a FTLA2000 spectrometer (ABB, Zürich, Switzerland) equipped with a MIRacle ATR sampling accessory (PIKE, Madison, WI, USA) in a set with a diamond crystal plate. Contact angles were determined using a NRL-100 contact angle goniometer (Rame Hart, Succasunna, NJ, USA) equipped with a tilting stage. Ellipsometric measurements were performed on a M-88 Variable Angle Spectroscopic Ellipsometer (J. A. Wollam, Lincoln, NE, USA) with a Hg-Xe laser (λ = 300−800 nm) at a fixed angle of incidence of 70°. The thicknesses of the layer underlying the PAA/P*t*BA films was determined experimentally based on the optical constants of these materials provided in the instrument software, and were then used to build a model. The PAA/P*t*BA film thickness, in a dry state, was determined using a Cauchy layer algorithm. Prior to this, we ensured that all carboxyl groups within the PAA brush were protonated by providing low pH conditions.

### 2.3. Immobilization of the Initiator on QCM Crystals

The QCM crystals were cleaned by immersion in 1% SDS solution, (DI) water, acetone and ethanol. At each time they were sonicated in respective medium for 5 min at constant temperature of 50 °C. They were then dried under a flow of gaseous nitrogen and exposed to air plasma (Harrick Plasma Cleaner, Ithaca, NY, USA) for 2 min to activate the surface [50]. Afterwards, the QCM crystals were immersed for 24 h at room temperature in solution containing 35 mL of ethanol and 80 μL of DTBU initiator. The initiator-grafted QCM crystals were rinsed with ethanol and sonicated in ethanol for 30 min at 25 °C.

### 2.4. ATRP Procedure for Tert-Butyl Acrylate Polymerization from Flat Substrates

CuBr_2_ (67.1 mg, 0.30 mmol), acetone (35.0 mL, 0.48 mol), PMDETA (64.7 μL, 0.31 mmol) and *tert*-butyl acrylate (12.0 mL, 82.7 mmol) were added to a 100 mL glass flask sealed with a rubber septum and degassed by purging with nitrogen for 1 h. Ascorbic acid (0.7 g, 4.0 mmol) was added to the mixture in a glovebox under an argon atmosphere and the solution was stirred (~5 min) until the color changed. For the grafting procedure initiator-grafted gold QCM crystals were dried under a flow of gaseous nitrogen and then transferred to the flask containing the polymerization solution. The polymerization was allowed to proceed at 25 °C for 22 h. Afterwards the QCM crystals were removed and rinsed with DI water, ethanol and dried under a flow of gaseous nitrogen.

### 2.5. Hydrolysis of Poly(tert-butyl acrylate) to Poly(acrylic acid)

The poly(*tert*-butyl acrylate) brush on a QCM crystal was converted into a poly(acrylic acid) brush by acid hydrolysis in dichlorometane (35 mL, 0.55 mol) and trifluoroacetic acid (3.0 mL, 39.2 mmol) according to established procedures [51,52,53]. The conversion proceeded for 18 h at ~0 °C.

### 2.6. HSA Adsorption: Influence of Salt Concentration and pH

The QCM crystals were calibrated at constant pH (pH = 7.2) in the buffer solution containing 10 mM of 3-morpholinopropane-1-sulfonic acid (MOPS) buffer and 10 mM NaCl. From this point forward such buffer solutions characterized by pH = 7.2 and I = 20 mM will be called starting buffer. The calibrated crystals were then subjected to a 5 g/L HSA suspension in the matching buffer solution (10 mM MOPS and 10 mM NaCl) at controlled temperature of 25 °C and a flow rate of 50 µL/min. Conditions of temperature and flow rate are unified for all solution used in this study. Afterwards QCM crystals were rinsed with the starting buffer.

The influence of the salt concentration was studied by a step-wise increase of ionic strength to I = 50, 75, 100 and 120 mM adjusted by NaCl added to the buffer. Afterwards QCM crystals were rinsed with the starting buffer. After the step-wise increase of the ionic strength the QCM crystals were immersed in the buffer solution of the same pH and ionic strength conditions as at the beginning of the experiment.

In the studies regarding the influence of pH the QCM crystals were then rinsed by buffer solution of constant ionic strength (20 mM) but different pH (pH = 6.5 and 7.6, respectively). In the final step the QCM crystals were rinsed with the starting buffer. After pH change, the QCM crystals were immersed in the buffer solution of the same pH and ionic strength conditions (pH 7.2 and I = 20 mM) as at the beginning of the experiment.

### 2.7. Response of Protein-Free Brush to pH

The QCM crystals were at first calibrated in buffer solution with ionic strength of 120 mM and pH of 7.2. After calibration, crystals were rinsed with buffer solution of I = 20 mM and pH of 7.2. In the following steps QCM crystals were rinsed with buffer solutions with pH of 6.5 and 7.6 at constant ionic strength (I = 20 mM). In the final step crystals were rinsed with buffer solution of I = 20 mM and pH 7.2.

### 2.8. Data Analysis

*f* versus time and *D* versus time plots were generated using the fifth overtone. Similarly, the fifth overtone was used for the mass estimation (see Section 3.5). The data analysis was performed using Q-Tools software from Q-Sense. In order to determine the mass change, ∆*m*, upon brush response to changing ionic strength and pH conditions we used a linear relationship between gain/lost mass and the change in resonant frequency according to the Sauerbrey [54] equation:∆*m* = −*C_QCM_* ∙ (∆*f*/*n*)(1)
where ∆*m* is the mass area change (in ng∙cm^−2^), *C_QCM_* the mass sensitivity (17.7 ng∙cm^−2^ Hz^−1^ at oscillation frequency of 5 MHz, or overtone *n* = 1) and the *n* is the overtone number (1, 3, 5, …). There are two situations in which the Sauerbrey relation might not be fully appropriate [55,56,57,58]. Firstly, the presence of water coupled to the attached layer can result in an overestimation of mass [59]. Secondly, the viscoelastic adlayer gives rise to a compound resonator for which *∆f* is not directly proportional to ∆*m* [59]. The Sauerbrey equation implies that if the changes of the normalized frequency ∆*f_n_*/*n* do not exhibit a significant dependence on the harmonic number, a given layer is a rigid one. The experimental results of ∆*f_n_*/*n* show only a small dependence on the overtone number (see Appendix A) which suggests that the Sauerbrey relation properly describes the behavior of our adlayers. Moreover a maximum dissipation changes compared with frequency shifts (see Appendix A) validates the Sauerbrey relation for this case [35].

## 3. Results and Discussion

### 3.1. Synthesis of the Brush Layer

The experimental procedure for the polymerization of poly(*tert*-butyl acrylate) brush and subsequent conversion into a poly(acrylic acid) brush is illustrated in Scheme 1. The successful polymerization of the PtBA brush on the surface of gold-coated QCM crystals was confirmed by FT-IR spectroscopy, contact angle analysis and ellipsometry. The FT-IR spectrum (see Figure 2) contains the expected peaks at 1731 cm^−1^ (C=O stretch) [60] and 2973 cm^−1^ (asymmetric CH_3_ stretching vibration) [60] and a doublet at 1370/1395 cm^−1^ (symmetric methyl deformation mode), showing the presence of the tBA moiety.[60] The measured contact angle (85°) correlates well with the literature values for PtBA layer assembled at the surface of poly(styrene) substrate [61], whereas, the measured contact angle for bare gold electrode was 62°. From ellipsometry the dry thickness of the polymer film was found to be 18 nm (see Table 1). The PtBA chains were converted to PAA via acidic hydrolysis. This was achieved by immersing the samples in a solution containing 35 mL of dichloromethane and 3 mL of trifluoroacetic acid for 18 h in an ice bath (~0 °C).

After hydrolysis the polymer brush thickness in a dry state decreased to 8 nm while the measured static contact angle decreased to 16° (see Table 1). This proves the increased hydrophilicity due to the presence of the PAA layer. The drop in the brush thickness can be attributed to the removal of the bulky *tert-*butyl groups. A similar behavior was reported recently [62]. The FT-IR spectrum showed a broad peak at 3000–3500 cm ^−1^, a broadening of the peak at 1731 cm ^−1^ and the loss of the peaks associated with the pendant methyl groups (see Figure 1), thereby documenting the successful cleavage (85%) of the *t*BA moiety [60]. A similar procedure was used recently in a synthesis of poly(methacrylic acid) brushes from silica nanoparticles [63].

### 3.2. Determination of the Grafting Density

The grafting density *σ* of the PAA brush was determined from Equation (2):σ = *hρN_A_*/*M_n_*(2)

Here *h* is the dry polymer thickness, *ρ* the density of PAA (=1.1 g/cm^3^) [64], *N_A_* Avogadro’s number, and *M_n_* the polymer molecular weight. To estimate *M_n_* we used a systematic comparison by Wu et al. on PAA brushes anchored to a flat silicon wafer with variation of the grafting densities at several ionic strengths [26]. As a result we can predict the thickness of the wet PAA brush *H* as it should present the dry thickness multiplied by the factor of 6.5 ± 0.5 (see Table 2).

Assuming that each monomer unit is a bead with diameter of 0.25 nm we can determine the number of monomer units *N_m.u._* within a single PAA chain. Multiplying the molecular weight of a monomer unit *M_m.u_*_._ by *N_m.u._* we can estimate the molecular weight *M_n_* of a single grafted PAA chain. The grafting density was estimated to be *σ* = 0.35 ± 0.13 nm^−2^. Such high grafting densities were also reported for PAA brushes achieved by a similar polymerization procedure on a flat silicon surface [26] where PAA brushes with grafting density up to 0.85 nm^−2^ have been achieved. However, in the present case the molecular weight of the single grafted PAA chain as well as the grafting density is only estimated. The wet thickness *H* of the PAA brush does not represent the total length of the polyelectrolyte chain, therefore, the estimated *σ* refers to the maximal grafting density of the analyzed brush.

### 3.3. Protein Adsorption onto Planar PAA Brush: Influence of Salt Concentration and pH

We studied the conformational response of a PAA brush with a pre-adsorbed HSA layer as a function of increasing salt concentration and changing pH. The course of experiment is described in Section 2.6. We used a combination of established protocols [32,35] based on varying ionic strength and changing pH to probe the brush response. Combined with the analysis of protein binding it allows us to interpret the QCM results based on our recent calorimetric studies as well as to compare them with vast number of reports on protein interaction with polyelectrolyte brushes [24,28,32,35,57,65,66].

Protein adsorbed strongly onto a-like charged PAA brush as indicated by the large Δ*f* shift in step I (see Figure 3). Correspondingly, Δ*D* increased sharply at first but then, after reaching a maximum, dissipation started to decrease. This likely suggests that HSA at first accumulated on the top of the brush and then started to “migrate” toward the inside of the brush, making the brush more packed and stiffer [67]. This we conclude from a slowly decreasing Δ*D* which suggests the formation of an increasingly organized structure of the PAA brush as it is complexed by HSA. According to Bittrich et al. the observed long equilibration time in this step arises form constant incorporation of protein into the brush-protein layer [32]. The rinse of HSA suspension with starting buffer in step II resulted in a Δ*f* increase (decreasing mass) and a Δ*D* decrease and suggests the removal of the bulky HSA molecules that have accumulated on the surface of the PAA brush. As a result, the brush should become more dissipative but the loss of the viscous layer of proteins seems to be decisive in the overall Δ*D* decrease.

In steps III to VI, the ionic strength was increased from 20 to 50, 75, 100 and 120 mM in a step-wise fashion, at constant pH. We attribute the observed systematic Δ*f* increase to protein desorption and commensurate brush collapse. Furthermore, we explain the small Δ*D* increase by the additional loss of HSA from the brush due to the increase of the ion concentration in the bulk solution [24,32]. Compression of the brush, driven by increasing the ionic strength, can help to expel any weakly bound HSA, as shown by Wong et al. [68] As a result the PAA brush becomes more dissipative. Rinsing of the brush with starting buffer in step VII decreased the ionic strength from 120 to 20 mM. The resulting Δ*f* decrease and concomitant increase in Δ*D* can be attributed to brush swelling due to the lower ion concentration in the bulk solution [24,32]. Our results are in good agreement with similar studies of brush conformation in aqueous solution [28,35,66]. A predominant role of counterions in protein adsorption onto PAA brush upon increasing salt concentration was reported recently [30]. Importantly, the difference in Δ*f* values between steps II and VII clearly suggests that even after increasing the NaCl concentration there is still a significant amount of HSA bound within the PAA brush.

At step VIII experiments with changing pH start. Upon changing the pH from 7.2 to 6.5 at constant NaCl concentration, Δ*f* increases and Δ*D* decreases. We attribute these changes to brush collapse due to the less pronounced protonation of the carboxyl groups which allows them to form more O-H bonds [65]. As shown by Welsch [69], the *pK_a_* of the carboxyl groups of acrylic acid polymers within the brush can increase by two units compared to the one in solution (*pK_a_* = 4.25 [70]). This phenomenon, known as the polyelectrolyte effect, arises from the mutual interactions of the neighboring, charged residues within the polyelectrolyte brush [71]. Therefore, even a small decrease in pH can result in marked protonation of the brush functional groups. Δ*f* and Δ*D* shifts upon subsequent increase of pH in step IX (from 6.5 to 7.6) indicate the exact opposite effect to the one described above. Finally, upon rinsing the QCM crystals with starting buffer in step X we observe no Δ*f* change and only a small Δ*D* increase which indicates brush swelling. Importantly, there is no significant difference in the Δ*f* value between the initial and final states of the pH-affected experiments (steps VII and X, respectively). It suggests that upon changing the pH we indeed observe brush swelling/deswelling rather than protein desorption.

#### Influence of pH on the Swelling of the PAA Brush

To verify whether the brush response during the pH-change arises from the additional HSA desorption or mainly from the brush swelling/deswelling, we examined the response of a protein-free PAA brush as a function of pH. The outcome of this experiment is presented in Figure 4. The time course of the experiment is described in Section 2.7, and was designed to enable direct comparison with steps: VII, VIII, IX and X of the protein-adsorption experiment.

Step A shows the response of the PAA brush to decreasing ionic strength (from 120 to 20 mM) at constant pH of 7.2. We attribute the observed Δ*f* decrease and corresponding increase in Δ*D* to brush swelling due to a decreased ion concentration in the bulk solution [24,32,72]. Upon changing the pH from 7.2 to 6.5 at constant ionic strength of 20 mM in step B, we observed the expected collapse of the brush due to the more pronounced protonation of the carboxyl groups [65]. Decreasing Δ*f* and increasing Δ*D* in step C (upon pH change from 6.5 to 7.6) indicate brush swelling. The carboxyl groups in this step are balanced by counterions to a greater extent than at pH 7.2 due to their increased dissociation driven by slightly alkaline conditions. As a consequence, the brush is not fully stretched which we conclude by comparison to the Δ*f* and Δ*D* shifts in steps A and C. Our results compare well with those reported by Liu et al. [66].

In step D (upon changing the pH from 7.6 to 7.2) we observe a strong Δ*f* decrease along with increasing Δ*D* indicating brush swelling. Such a behavior was not observed between steps IX and X of the protein-adsorption experiment due to the presence of bound protein molecules within the PAA brush. The internal friction of a swellable polymer brush can be considerably increased by protein incorporation as shown by Bittrich et al. [32]. The following analysis confirms that in the case of the pH changes in the studies of protein adsorption (see Section 3.3) we mainly observe the swelling of protein-complexed PAA brush rather than the protein desorption.

### 3.4. The Amount of Adsorbed Protein Is Determined by the Ionic Strength

We used the Sauerbrey equation (Equation (1)) [54] to extract the changes of the mass density (Δ*m*) of the brush at each step of the experiments described above (see Figure 5) to estimate the amount of HSA adsorbed per PAA chain. The observed brush response in steps VIII, IX and X corresponds to the swelling/deswelling of the PAA brush induced by changing pH. The calculated values of Δ*m* upon ionic strength- and pH- changes as well as the Δ*m* values upon pH induced swelling of a protein-free brush are presented in Table 3, Table 4 and Table 5, respectively.

The changes of the mass density between each step of the pH-influence experiment (see Figure 5a) compared well to the changes of the mass density between analogous steps of the pH induced swelling of the protein-free PAA brush (see Figure 5b), indicating that protein desorption during the pH-change is marginal. Consequently, during the pH-change we observe swelling of the protein-complexed brush. From the difference in the changes of the mass density recorded upon swelling of the protein-free brush (Table 5) we can estimate the effect of the coupled solvent in our experiments. The largest ∆*f* shift can be observed between steps A and B thus the coupled solvent can be expressed as ± 49 Da/A^2^. In this way we can correct our results to better determine the amount of HSA adsorbed per PAA chain and lost during increasing the ionic strength.

The difference in Δ*m* between the initial and the final steps of increasing ionic strength (steps II and VII in Table 3) is of about 118 ± 49 Da/A^2^. This reflects the amount of desorbed HSA, and thus highlights the major influence of counterions in the process of polyelectrolyte mediated protein adsorption/desorption [73,74,75]. The remaining 220 ± 49 Da/A^2^ can therefore be attributed to the HSA molecules that are attached to the PAA brush with higher affinity than the proteins desorbed during the increase of the ionic strength. This might indicate the existence of two fractions of HSA molecules within the PAA brush: those with high- and low binding affinity (see Figure 6).

The presence of high- and low binding affinity sites for proteins within a polyelectrolyte brush was previously observed for β-lactoglobulin binding onto spherical polyelectrolyte brushes (SPBs) [76]. Here, due to the planar geometry of our polyelectrolyte brush, the presence of high- and low binding affinity sites can be attributed to the polydispersity of the brush chain. While at high grafting density occurs close to the surface, the chain segment density decreases towards the distal end of the brush (see Figure 6).

The presence of a HSA fraction with low binding affinity indicates that this phenomenon is an example of a negative cooperativity. Therefore, high- and low binding affinity may reflect the difference in protein adsorption onto free- and already preoccupied polyelectrolyte chains. A similar result was reported recently in the case of HSA adsorption onto PAA-based spherical polyelectrolyte brushes (SPB) [30].

### 3.5. Number of HSA Molecules per PAA Chain

From the change of the mass density at Step II (the initial state of the increasing ionic strength) with correction for the effect of coupled solvent and from the molecular weight of HSA (see Section 2.1) we can determine the number of HSA molecules per nm^2^ (see Table 6). Comparing this number to the inverse grafting density *σ*^−1^ = 2.9 ± 0.5 nm^2^ of PAA brush allows to estimate the amount of HSA initially adsorbed per PAA chain.

In the same way we can verify that at Step X (the final state of pH-change) approximately one HSA molecule is adsorbed per one PAA chain (see Table 6)—the evaluation of those data are presented in the Appendix A. Therefore, approximately 40% of initially adsorbed HSA molecules are desorbed during the increase of the ionic strength.

These numbers are consistent with results reported by Yu et al. who used a combination of calorimetry and computer simulation to verify that HSA binds with a single short PAA chain [36]. Our results agree also with work of Wittemann et al. in which bovine serum albumin (BSA) molecules were released from PAA-based SPBs by washing them off with solution of higher ionic strength [77]. They reduced the number of attached BSA molecules from two per PAA chain to about one per two PAA chains. In our recent work of HSA adsorption onto SPBs we also verified the influence of ionic strength on the binding [30]. With increasing salt concentration the repulsion between the protein and polyelectrolyte brush become operative thus suppressing the adsorption process.

## 4. Conclusions

In the present study, we have successfully demonstrated the ARGET ATRP polymerization of PAA brushes grafted from a planar gold surface of QCM crystals. The adsorption of HSA onto and the desorption from PAA brush as a function of ionic strength and pH was investigated by QCM-D. Conformational changes of the PAA brush were observed and used to correct the values measured for HSA adsorption. By releasing a part of initially adsorbed protein molecules upon increasing salt concentration we demonstrated the dominant role of counterions in the process of polyelectrolyte mediated protein adsorption/desorption. By comparison with recent calorimetric studies on the protein interaction with polyelectrolytes [30,36] we presented a new approach in which QCM data are analyzed based on the results of thermodynamic analysis.

Finally, we conclude that QCM crystals modified through presented method based on ARGET ATRP reaction are fully functional. A comparison with large number of other brush systems interacting with proteins [21,29,30,36,77] lead to a full agreement in the number of adsorbed protein molecules per polyelectrolyte chain at low and high ionic strength. Thus, the present findings extend our understanding of interaction between protein and polyelectrolyte brush by comparison of systematic studies of protein adsorption/desorption driven by increased salt concentration with calorimetric studies of the same problem.

## Data Availability

The data presented in this study are available within the article and Appendix A.

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
