# Peer review of "Interaction of Proteins with a Planar Poly(acrylic acid) Brush: Analysis by Quartz Crystal Microbalance with Dissipation Monitoring (QCM-D)"

_polymers, 2020, doi:10.3390/polym13010122_

Round 1

Reviewer 1 Report

This paper describes the effects of pH and ionic strength of media on the interaction of PAA brushes with HSA by using QCM-D. The results are well discussed based on the findings reported previously, which supports the conclusions. The interaction of polymer brushes with proteins in aqueous media would also be influenced by the thickness and grafting density of PAA brushes. Such additional studies might be useful to understand the insight of the interaction of PAA with HAS by combining with the present results. This is just a comment for future study.

I think that the authors should add a clear answer about the reason why the decrease in Δf and the increase in ΔD, indicating the swelling PAA brushes, were observed when the pH of media was changed from pH 7.6 to 7.2. (Figure 4, step D) I could understand why the distinct QCM-D response was observed by such a slight change in pH.

Reviewer 2 Report

I believe that the authors have prepared and presented a very well written manuscript. Their work involves an extented characterization to the system presented. I only have some minor corrections and some questions regarding their experimetal part.
